# Malignant Giant Cell Tumor of Bone: A Study of Clinical, Pathological, and Prognostic Profile from One Single Center

**DOI:** 10.3390/bioengineering12090911

**Published:** 2025-08-25

**Authors:** Jingtian Shi, Xin Sun, Jichuan Wang, Haijie Liang, Xingyu Liu, Yi Yang, Xiaodong Tang, Wei Guo

**Affiliations:** Musculoskeletal Tumor Center, Peking University People’s Hospital, Beijing 100044, China; sjt_1996@163.com (J.S.); xinsun1981@hotmail.com (X.S.); lianghaijie_glia@163.com (H.L.); lxysunknight@bjmu.edu.cn (X.L.); 13701312827@163.com (Y.Y.); tang15872@163.com (X.T.)

**Keywords:** malignant GCTB, primary, secondary, axial bone, pelvis, *H3F3A*, chemotherapy, denosumab

## Abstract

Malignant giant cell tumor of bone (GCTB) is a rare malignant bone tumor. This analysis was conducted on patients with malignant GCTB at our center. The clinical, demographic, and prognostic characteristics were evaluated and compared. During 1 January 2015 to 31 December 2022, fifty patients were included in the study, which made up 3.3% of the contemporary GCTB patients. The clinical characteristics were comparable between the 24 patients with primary malignant GCTB (PMGCTB) and 26 patients with secondary malignant GCTB (SMGTCB). The tumor location was mainly at the axial and pelvic region (70%) and differed between the two types (*p* = 0.040). *H3F3A* pathogenic variant presented frequently in SMGCTB (*p* = 0.020). Cox regression analysis showed the prognostic outcomes were poor in those with a tumor located in the axial bone and sacrum with invasion of other places. *H3F3A* mutation status is also a risk factor, while chemotherapy and denosumab failed to demonstrate prognostic benefits. Malignant GCTB is a rare condition with a poor prognosis, especially in SMGTCB. The location and *H3F3A* mutation status had an influence on prognosis, and systemic therapy should be taken into consideration for patients with unfavorable prognostic features.

## 1. Introduction

Giant cell tumor of bone (GCTB) is a locally aggressive benign bone tumor. It is mainly composed of three components, including macrophage-like round cells, “reactive” osteoclast-like multinucleated giant cells, and “neoplastic” fibroblast-like spindle stromal cells [1,2,3]. GCTB can occur anywhere along the skeleton, while it mostly affects the epiphyseal and metaphyseal regions of long bones and occasionally in the axial bones (spine and pelvis) [4,5,6,7]. The most likely primary driver of the tumoral cell appears to be a mutation in *H3F3A*, leading to a substitution of glycine (Gly)34 to tryptophan (Trp) in histone H3.3. This change presumably alters the methylation of the protein, and thus, its effect on gene expression [8,9]. Generally, radical surgery remains the primary treatment for most cases. When the surgery is not possible or it would be associated with excessive morbidity, other adjuvant treatments would be taken into consideration, such as radiotherapy [10,11,12], embolization [13,14], and drugs [15,16,17]. The elucidation of the biology of GCTB led to trials of denosumab in this disease, which is a monoclonal antibody of receptor activator of nuclear factor-κB ligand (RANKL) expressed by the stromal cells. In selected cases with GCTB, denosumab has emerged as an effective first-line treatment [15,18].

Although the majority of GCTBs present with a benign feature, they can appear as or transform into a malignant one. The term “malignant GCTB” first appeared in the 1930s [19] and was used to describe a sarcoma arising from a giant cell tumor [20]. Malignant GCTB is a rare sarcoma, with the incidence estimated to be 2–15% in patients with benign GCTB [10,21,22,23,24]. Based on the WHO classification, it could be divided into two subtypes, namely, the primary malignant GCTB (PMGCTB) and secondary malignant GCTB (SMGCTB) [25]. The PMGCTB refers to the cases of benign GCTB juxtaposed with high-grade sarcoma. And the SMGCTB is recognized in which the sarcoma occurs at the site of previously managed GCTB, usually after surgery, radiation, or both. In a recent review, reported overall incidence of PMGCTB in patients with GCTB was 1.6% and 2.4% for SMGCTB, and both types displayed a poor outcome [26]. Previous studies reported that the *H3F3A* mutation status remained positive in malignant GCTB [27,28]. Malignant transformation of GCTB can cause great disruption of bone and even lead to more aggressive lung metastasis, which results in a relatively poor prognosis; thus, it is important to make a correct diagnosis to facilitate appropriate treatment.

Previous studies have reported information on MGCTB, while most of their cases consist only of limb lesions. No current studies including a relatively large number of cases have reported the prognosis of patients with malignant GCTB. Therefore, we aim to investigate the clinicopathological characteristics and prognostic outcome of patients with malignant GCTB based on our single-center data. The following objectives are to be ascertained: 1. the clinical and demographic characteristics of patients with malignant GCTB; 2. the clinical outcomes of these patients; 3. potential factors on the prognosis; 4. if the patients could derive benefits from systemic therapy.

## 2. Materials and Methods

### 2.1. Data Source and Inclusion Criteria

We reviewed all cases from our single institute in which the diagnosis of GCTB was related synchronously or metachronously to a diagnosis of sarcoma from 1 January 2015 to 31 December 2022. The inclusion criteria were as follows: 1. patients with malignant GCTB diagnosed by histology and received treatment at our center; 2. the demographic, clinical, and pathological features, and prognostic outcome could be obtained; 3. with at least a 12-month follow-up or those who died with a complete follow-up less than 12 months. The exclusion criteria were as follows: 1. patients with diagnosis of giant cell-rich osteosarcoma; 2. patients who received only systemic therapy instead of surgical resection; 3. patients with incomplete follow-up information or those with follow-up less than 12 months. All data in our study were obtained from a database that was set for searching clinical data regarding patient demographics, clinicopathological features, radiological examination, and cancer-associated treatment. This study was approved by our center’s Institutional Review Board, which dispensed with the need for informed consent, as our study was a retrospective and minimal risk one, and any identifiable personal information was not collected. All the methods and research were performed in accordance with the relevant guidance and regulations of the ethics commission.

### 2.2. Definition of Variables

Clinical, demographic, and pathological information was collected and analyzed accordingly. Data regarding gender, age at presentation, tumor type, location, denosumab management, potential cause, previous treatment and latency period of SMGCTB, mutation status of *H3F3A* gene, chemotherapy, local recurrence, lung metastasis, survival state with the associated period, and follow-up duration were reviewed. The tumor type was divided into primary and secondary. Patients were divided simply by denosumab management into two groups: those who received denosumab therapy and those who did not. The potential cause of SMGCTB was considered as surgery-induced and radiation-induced. Some of the patients underwent both the treatment, and we categorized these cases as the latter form, for which the radiation was a prominent factor in inducing malignant transformation in GCTB. The latency period in cases of SMGCTB was defined as the interval between the treatment of benign GCTB and the diagnosis of malignant transformation to high-grade sarcoma. The location of the tumor was divided as limb, limb girdle bone (pelvis and scapula), axial bone (spine and sacrum), and others (with simultaneous involvement of sacrum and lumbar vertebrae or sacrum and pelvis). Previous studies have demonstrated that approximately 85% to 95% of GCTB harbor the *H3F3A* G34W pathogenic variant, and a minor subset have G34V and G34R [29]. So, in the present study, the staining of the *H3F3A* gene on G34W, G34V, and G34R was performed. For the 29 specimens obtained in the early time, re-staining for the potential *H3F3A* pathogenic variant is necessary. However, the staining results were still unavailable for 5 specimens in this cohort. The slides were stained and reviewed again by an experienced pathologist (SKK). *H3F3A* mutation status was defined as positive when staining for one of the above-mentioned proteins was positive, while negative when all the pathogenic variants were negative. The chemotherapy regimen was mainly composed of cisplatin, doxorubicin, high-dose methotrexate, and ifosfamide, a protocol for osteosarcoma. The length of follow-up was calculated from the curative surgery performed at our institution until the last date of evaluation or death.

### 2.3. Statistical Analysis

The parametric variables are described by mean with standard deviation (M ± SD) in those with normal distribution and median with interquartile range (Med, IQR) in non-normal distribution ones, while the categorical variables are presented with frequencies with proportions (N, %). Student *t*-test or ANOVA analysis was performed to examine significance among the continuous variables when appropriate. Chi-square test was performed for categorical variables. Nonparametric test (Mann–Whitney U test) was performed among variables with non-normal distribution. The local recurrence-free survival (RFS), distant metastasis-free survival (MFS), and overall survival (OAS) were evaluated based on the intervals from the time of initial surgery to the time of the first local recurrence, the first distant metastasis, or death, and analyzed using the Kaplan–Meier curve, respectively. The patients with distant metastasis at the time of diagnosis were not included in the calculation and labeled as contemporary metastasis. Univariate Cox hazard regression was used for analysis to determine the factors that were predictors for RFS, MFS, and OAS, and compared by the log-rank test. A *p*-value of less than 0.05 was considered statistically significant. All the statistical analyses were performed using Stata 15 for Windows (Stata Corporation, College Station, TX, USA).

## 3. Results

### 3.1. General Characteristics

The clinical and demographic characteristics of patients are presented in detail in Table 1. During the same period, a total of 1520 patients were diagnosed as GCTB (either benign or malignant) at our center, of whom 50 patients (3.3%) were treated and histologically confirmed as malignant GCTB. All the patients received marginal or wide resection according to the location of the lesion (Figure 1). The mean age of the patients was 45.7 years, and there was an equal gender distribution of the included patients (1:1). The most frequent location of the tumor was the limb girdle bone and axial bone, which accounts for 70% of the patients. Twenty-four patients had some type of PMGCTB, and 26 had SMGCTB, of which 19 patients (73.1%) were secondary to surgery. Of these patients secondary to surgery, initial surgery was performed with resection in eight patients, while most had malignant transformation after curettage (11, 57.9%). The malignant component of the lesion was mainly divided into five types, and the OS and UPS took up 38.0% and 39.0%, respectively. The *H3F3A* mutation status was positive in 66.7% of the patients. The most common substitution is G34W, with rare variants of G34V and G34R. Lung metastasis at presentation was identified in merely three patients (6%).

### 3.2. Comparison of the Patients with PMGCTB and SMGCTB

There was no difference with regard to the gender distribution or the age between the patients with PMGCTB and SMGCTB (Table 2). When it comes to the location of the tumor, patients with PMGCTB tended to arise from the pelvis (45.9%) while axial bone in the SMGCTB (57.7%). In addition, the malignant component of the lesion was most frequently identified as UPS (45.8%) in patients with PMGCTB and OS (46.2%) in SMGCTB, although the difference was not significant (*p* > 0.05). The *H3F3A* mutation status was most frequently positive in patients with SMGCTB (82.6% vs. 50.0%, *p* = 0.020) and H3.3G34V substitution (17.4% vs. 0, *p* = 0.040). Denosumab was administered to most patients with SMGCTB compared to PMGCTB (69.2% vs. 41.7%, *p* = 0.050), while the proportion of patients who received chemotherapy was slightly higher in patients with PMGCTB (37.5% vs. 15.4%, *p* = 0.075). In patients with SMGCTB, there was no difference in the clinical characteristics between those secondary to surgery and radiotherapy (Table 3). Noticeably, the median latency interval was slightly longer in patients secondary to radiotherapy (90 vs. 31 months, *p* = 0.140).

### 3.3. Clinical Outcomes and Potential Risk Factors for Prognosis

The estimated 5-year RFS, MFS, and OAS were 49.1% (95%CI, 33.6–62.9%), 62.2% (95%CI, 45.0–75.4%), and 58.7% (95%CI, 42.3–71.8%) for all the included patients, respectively. In 26 patients with PMGCTB, 9 patients (37.5%) developed local recurrence during a median of 8 (IQR, 3–8) months. In contrast, there were 15 cases (57.7%) of recurrence observed in 24 patients with SMGCTB during a median of 10 (IQR, 5–19) months. Local recurrence of SMGCTB occurred in 71.4% (5 of 7) patients secondary to radiotherapy during a median of 10 (IQR, 6–19) months, while only 52.6% (10 of 19) patients secondary to surgery recurred (*p* = 0.390) during a median of 11 (IQR, 5–12) months. Excluding the patients with metastasis at presentation (n = 3), 10 of 24 patients (41.7%) had lung metastasis in SMGTCB during a median of 10 (IQR, 5–21) months, whereas 6 of 23 patients (26.1%) had metastasis during a median of 5 (IQR, 3–10) months. The incidence of lung metastasis was comparable between patients with SMGCTB, as well as the interval of occurrence of metastasis. At the last follow-up, a total of 70.8% (17 of 24) patients with PMGCTB were still alive, while only 53.8 (14 of 26) patients with SMGCTB survived (*p* = 0.216). However, more than half of the patients with SMGCTB secondary to surgery were alive, a relatively higher incidence when compared to those secondary to radiotherapy (57.9% vs. 42.9%, *p* = 0.495). The median time of survival was slightly longer in patients with SMGCTB secondary to radiotherapy (19 vs. 12 months), which was not seen in comparison between patients with PMGCTB and SMGCTB (12 vs. 12.5 months). The outcomes of patients with specific *H3F3A* pathogenic variants are displayed in Table 4.

Upon univariate analysis (Table 5), patients who had the tumor located at the axial bone and with simultaneous involvement of the pelvic and sacrum presented a higher risk of local recurrence (Figure 2) when compared with patients who had the tumor located at the limb (HR:11.06 [95%CI: 1.44–85.76], *p* = 0.021; HR: 56.73 [95%CI: 5.87–548.51], *p* < 0.001). No other variables could influence the incidence of recurrence. In terms of lung metastasis, location and *H3F3A* mutation status, especially the H3.3G34W pathogenic variant, were independent predictors for MFS (Table 5, Figure 2). Patients with the tumor involving the pelvis and sacrum and those without *H3F3A* pathogenic variants tended to have a higher incidence of lung metastasis during follow-up (HR:15.31 [95%CI: 1.35–173.37], *p* = 0.028; HR: 3.16 [95%CI: 1.10–9.14], *p* = 0.033). In patients with positive status of *H3F3A* mutation, the H3.3G34W pathogenic variant is also a risk factor for MFS, with those negative presenting a higher risk of lung metastasis (HR:8.87 [95%CI: 1.97–40.05], *p* < 0.001). As shown in Table 5 and Figure 2, location and H3.3G34R pathogenic variant were significantly associated with OAS (HR:8.20 [95%CI: 1.06–63.61 for axial bone, *p* = 0.044; HR: 17.90 [95%CI: 1.98–162.14] for pelvis and sacrum, *p* = 0.010; HR: 0.26 [0.07–0.94] for H3.3G34R positive, *p* = 0.04). No statistical treatment benefits were found in the included patients with chemotherapy and denosumab for RFS, MFS, and OAS (*p* > 0.05, Figure 3).

## 4. Discussion

GCTB is a locally aggressive benign bone tumor with most of the cases affecting the knee joint [20,21,30]. Sometimes the malignancy of GCTB could be noticed. In the current study, the malignancy of GCTB accounted for 3.3% of all GCTB patients, which is in accordance with previous studies [10,21,31]. Given the increasing multi-modality treatment in malignant tumors, a better understanding of the characteristics and behavior of the malignant GCTB could help predict the prognosis and identify the patients who could respond to systemic therapy. Therefore, in the current study, we aimed to present an update on clinical, pathological, and prognostic factors of patients with malignant GCTB based on a review of patients in our single center and draw a clearer understanding of the treatment and prognosis of this rare disease to manage it better.

### 4.1. Clinical, Demographic, and Prognostic Differences

The clinical and demographic features of patients with PMGCTB and SMGCTB were similar except for the tumor location, treatment history, *H3F3A* mutation status, and the application of denosumab. The tumor location in this study included the pelvis for most of PMGCTB, while it was in axial bone for SMGCTB, which illustrated that the GCTB in axial bone is more likely to experience intralesional curettage or radiotherapy due to anatomical constraints. While most of the patients with PMGCTB in our center had lesions in the pelvis, patients with SMGCTB had a significantly higher incidence of multiple surgeries after malignant transformation of the primary benign lesion. More than 90% of the patients came to ask for help from us with recurrent SMGCTB, which caused a relatively poor prognosis of this entity. In addition, the *H3F3A* mutation status had a higher positive rate in those with SMGCTB. It could not be further determined whether this represents an acquired mutation after malignant transformation or inherent phenotype of the origin lesion, because of lack of the initial pathology. Denosumab is a common regimen used in GCTB to downstage, and thus most of the patients with SMGCTB received denosumab for local control of prior benign GCTB. No other difference was observed between patients with PMGCTB and SMGCTB, as well as those with SMGCTB secondary to surgery and radiotherapy. While noteworthy, our findings revealed that the median latency interval between the treatment to the malignant transformation was longer in patients with SMGTCB secondary to radiotherapy (90 vs. 31 months, *p* = 0.140). The prognostic outcome was relatively poor in SMGCTB (Table 2), especially those with malignancy secondary to radiotherapy (Table 3).

### 4.2. Prognosis Influenced by the Tumor Location

The anatomical location of malignant GCTB in this study was different from that observed by other authors [21,32], as most of the patients had tumors occurring at limbs, while at axial bone and pelvis in ours. Our findings showed that the lesion arising from the axial bone had a worse RFS, MFS, and OAS. In general, the size of lesions in this site is often relatively larger than those in the extremities. And there are many important neurovascular bundles and organs surrounding this place, making marginal excision difficult to perform. In addition, massive intraoperative hemorrhage reduces the definition of the field of surgery. The sacroiliac joint (SJI) cartilage or L5 disk acts as a barrier to tumor invasion of the joint space; the involvement may reflect a high capacity for the tumor to invade. All these could lead to worse local control and prognosis. However, in the included patients of this study, the malignant GCTB located in the pelvis did not present a poor outcome when compared to those in the extremities (Table 5). This may be due to the fact that most patients with pelvic tumors had PMGTCB (11/15, 73.3%), and thus, wide resection could be performed when compared to SMGCTB, which underwent multiple surgeries or radiation. In addition, the abundant experience of treatment of the pelvic tumor in our center led to this result, which is comparable to other pelvic malignant tumors of previous studies [33,34,35,36].

### 4.3. H3F3A Mutation Status and Pathogenic Variants

Several recent studies have demonstrated that *H3F3A* mutation status might contribute to distinguishing GCTB-related tumors. The *H3F3A* mutation status, detected with immunohistochemical or molecular methods, is usually retained in the malignant population [29], but there are cases where it is lost instead [37]. There is no known specific genetic signature of malignant GCTB. Therefore, malignant bone sarcomas with H3.3G34 pathogenic variant should be considered as malignant GCTB [29]. In our study, the *H3F3A* mutation status was detected as positive in 66.7% of the patients, with the H3.3G34W pathogenic variant being the most common in our cases (24/45, 53.3%). In addition, we found the *H3F3A* mutation status to be a further independent risk factor for MFS. However, this was not seen in RFS and OAS. Patients without the *H3F3A* pathogenic variants are associated with a high risk of lung metastasis, which means those malignancies lacking of *H3F3A* driver mutation present a more aggressive behavior. However, we could not further obtain the specimens of lung metastasis to confirm the *H3F3A* mutation status. For patients with the *H3F3A* pathogenic variant on G34W, the same phenomenon was observed when compared to those without H3.G34W. Patients exhibiting expression of the H3.3G34R pathogenic variant demonstrated significantly inferior OAS when compared to their H3.3G34R-negative counterparts. The potential mechanism of this situation could not be identified. Thus, we advocate regular and strict monitoring and aggressive treatment for patients with H3.G34W negativity and H3.G34R positivity.

### 4.4. Treatment Benefit with Chemotherapy and Denosumab

The role of chemotherapy for malignant GCTB remains controversial, and there is still insufficient evidence for the benefit of oncological outcome. Antract reported a better 1-year survival rate in patients who were treated by a combination of surgery and chemotherapy when compared to those who underwent surgery alone [24], while this disappeared in the 5-year survival rate comparison. Another study showed that adjuvant chemotherapy as a salvage treatment following surgery with an inadequate margin did not result in any obvious advantage [10]. In contrast to findings of former studies, several studies have reported that the use of chemotherapy offered some benefits [31,38]. Liu et al. have observed that chemotherapy benefited MFS, in which the 5-year survival rates were 57.0% and 33.3% in chemotherapy and non-chemotherapy groups, respectively [21]. In the current study, no significant survival advantage was observed with chemotherapy. While chemotherapy was applied only in 17 patients, these patients presented with multiple confounding factors in terms of tumor type, location, and treatment history. This made it challenging to directly evaluate the therapeutic role of chemotherapy. Thus, even though our data did not support the efficacy of chemotherapy for malignant GCTB, we should interpret the results cautiously. Given that the number of included patients is small, the therapeutic role of chemotherapy in malignant GCTB remains inconclusive. Given the outstanding results achieved by chemotherapy in other malignant bone tumors, we suggest administration of chemotherapy for selected patients with malignant GCTB, especially for those with the inferior prognostic features mentioned above.

In recent years, with the increasing application and promising results of denosumab as adjuvant therapy in patients with unresectable benign GCTB or those with planned surgery expected to result in severe morbidity [18,39], we wonder if denosumab could also provide prognostic advantages in patients with malignant GCTB. Although in the setting of denosumab therapy, malignant transformation has been sometimes reported to arise [15,40], this phenomenon is sufficiently uncommon such that it remains unclear whether denosumab might be causative [41], as only 1% patients on denosumab therapy have experienced malignant transformation, similar to the prevalence of malignant transformation in denosumab-naïve tumors. Overall, denosumab is a safe and effective therapy for GCTB. Denosumab is a full human monoclonal antibody specifically targeting RANKL, a key regulator of osteoclast formation and function. By binding to RANKL with high affinity, denosumab prevents its interaction with the RANK receptor on osteoclast-like giant cells, which leads to bone destruction in conditions of GCTB [18,42]. This inhibition blocks osteoclast-like giant cell-mediated bone resorption and reduces bone destruction. Additionally, it disrupts the tumor-induced osteolytic cycle, slowing disease progression and promoting bone repair. Preoperative denosumab could lead to new ossification and reappearance of cortical integrity around the soft tissue mass, which helps to identify the tumor margin when performing wide resection or convert it to intralesional curettage [43]. Together, a decreased vascularity is expected to minimize tumor contamination and facilitate complete surgery [44]. However, a study revealed that the malignant GCTB lacks response to neoadjuvant denosumab treatment, although they did not compare the RFS, MFS, and OAS due to small sample size [45]. In the current study, the efficacy of denosumab administration was also not statistically prominent in terms of RFS, MFS, and OAS (Figure 3). This situation may be due to the fact that most patients in this study who received denosumab were SMGCTB (18 of 28, 64.3%), and majority of tumors were located in the axial bone (11 of 18, 61.1%). Local fibrosis formation and obscured anatomical planes caused by malignant transformation after curettage or radiotherapy further contributed to the significant challenge of obtaining adequate surgical margins in subsequent resections. As a result, these patients did not receive benefits from denosumab treatment in terms of local tumor control, lung metastasis prevention, and long-term survival. Nevertheless, the reliability of our results may have been lessened because of the bias in patient selection. Therefore, additional prospective studies are warranted to evaluate the efficacy of denosumab in the management of malignant GCTB.

### 4.5. Limitations

Some limitations should be considered when interpreting our results. First, all available data in the current study is obtained in a retrospective way; therefore, this investigation has the inherent constraints of a retrospective design. Second, the small number of cases might render the study underpowered to detect the treatment benefits. In addition, some detailed information about the treatments is missing, such as the administration of denosumab. More specifically, data regarding the duration and frequency of denosumab administration were not available. However, in spite of the above-mentioned limitations, the current study has, as we know, the largest number of patients with malignant GCTB reported to date, and most of the lesions are located in axial bone and pelvis, which are surgical sites with complex anatomy. In addition, all the surgeons in our center performed the treatment according to the current guidelines for malignant tumors. Therefore, the margin of surgical intervention was well defined, which could make our study results less susceptible to measurement errors in outcomes. We believe that the present article could provide an objective evaluation and comprehensive information on patients with malignant GCTB, and create a good basis for evidence-based decision making.

## 5. Conclusions

In our study, we found that the patients with malignant GCTB exhibited dramatically poor prognosis, especially for SMGCTB. The prognosis was mainly correlated with the tumor location, *H3F3A* mutation status, and its pathogenic variants. The systematic treatment of chemotherapy and denosumab showed no benefits for RFS, MFS, and OAS, while we still advocate systemic treatment of chemotherapy and denosumab for those with poor prognostic features.

## Figures and Tables

**Figure 1 bioengineering-12-00911-f001:**
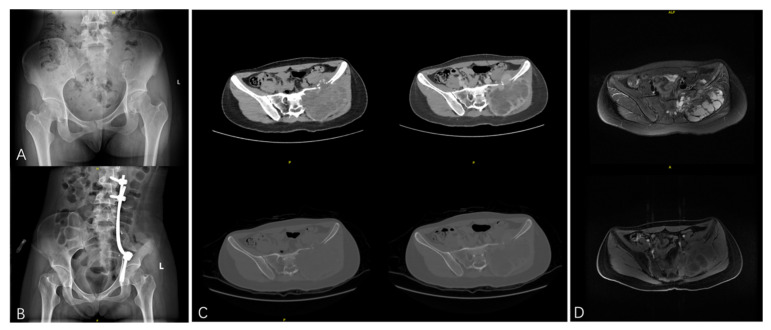
PMGCTB of the left pelvic and with invasion to the sacrum treated with marginal resection: (**A**) A 20-year-old female presented with persistent pain in the left buttock. X-ray examination showed bone destruction of the left ilium. (**B**) The postoperative X-ray of the patient. (**C**) Manifestations in plain CT and contrast-enhanced CT before surgery. Plain CT showed an osteolytic lesion in the left ilium with a large soft tissue mass. The ipsilateral sacrum was also invaded. Contrast-enhanced CT demonstrates heterogeneous enhancement of the tumor. (**D**) Manifestations in T2WI of plain MRI and T1WI of contrast-enhanced MRI before surgery. MRI revealed a multiloculated cystic mass with fluid-fluid levels. The lesion was heterogeneously hyper-intense in T2WI and showed enhancement in contrast-enhanced MRI.

**Figure 2 bioengineering-12-00911-f002:**
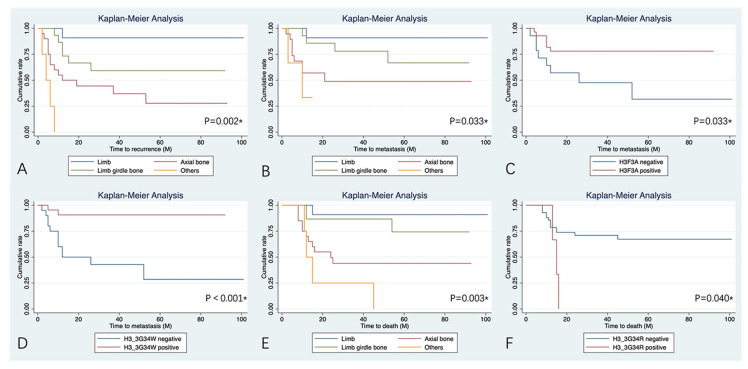
Kaplan–Meier curves of the factors that influence the prognosis: (**A**) The RFS for different locations of the tumor. (**B**) The MFS for different locations of the tumor. (**C**) The MFS for the *H3F3A* staining of the tumor. (**D**) The MFS for the staining of the *H3F3A* pathogenic variant H3.3G34W of the tumor. (**E**) The OAS for the different locations of the tumor. (**F**) The OAS for the staining of the *H3F3A* pathogenic variant H3.3G34R of the tumor. * means the *p*-Value is significant.

**Figure 3 bioengineering-12-00911-f003:**
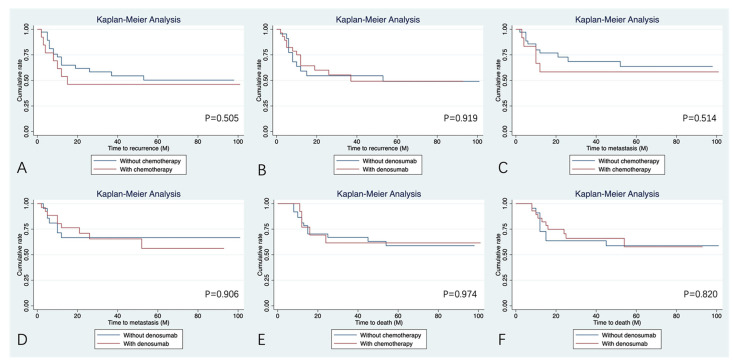
Kaplan–Meier curves of the systemic treatment (chemotherapy and denosumab) for the prognosis. (**A**,**C**,**E**) Chemotherapy shows no benefits for the RFS, MFS, and OAS. (**B**,**D**,**F**) Denosumab shows no benefits for the RFS, MFS, and OAS.

**Table 1 bioengineering-12-00911-t001:** The characteristics of the patients included in the study.

Characteristics	Value
Gender (N, %)	
Male	25 (50.0%)
Female	25 (50.0%)
Age (Year, Mean ± SD)	45.7 ± 14.9
Location (N, %)	
Limb	11 (22.0%)
Limb girdle bone	20 (40.0%)
Axial bone	15 (30.0%)
Others	4 (8.0%)
Treatment history (N, %)	
Newly	24 (48.0%)
Recurrent	26 (52.0%)
Type (N, %)	
Primary	24 (48.0%)
Secondary	26 (52.0%)
Secondary to (N, %)	n = 26
Surgery	19 (73.1%)
Radiotherapy	7 (26.9%)
Surgery history (N, %)	n = 19
Curettage	11 (57.9%)
Resection	8 (42.1%)
Pathology (N, %)	
OS	18 (36.0%)
CS	4 (8.0%)
FS	6 (12.0%)
UPS	19 (38.0%)
MFH	3 (6.0%)
*H3F3A* mutation status (N, %)	n = 45
Positive	30 (66.7%)
Negative	15 (33.3%)
H3.3G34W (N, %)	n = 45
Positive	24 (53.3%)
Negative	21 (46.7%)
H3.3G34V (N, %)	n = 45
Positive	4 (8.9%)
Negative	41 (91.1%)
H3.3G34R (N, %)	n = 45
Positive	3 (6.7%)
Negative	42 (93.3%)
Initial metastasis (N, %)	
Yes	3 (6.0%)
No	47 (94.0%)
Denosumab (N, %)	
Yes	28 (56.0%)
No	22 (44.0%)
Chemotherapy (N, %)	
Yes	13 (26.0%)
No	37 (74.0%)
Follow-up (Mons, Mean ± SD)	41.6 ± 29.3

OS: osteosarcoma; CS: chondrosarcoma; FS: fibrosarcoma; UPS: undifferentiated pleomorphic sarcoma; MFH: malignant fibrous histiocytoma.

**Table 2 bioengineering-12-00911-t002:** Comparison of the characteristics and prognosis according to the different types.

Characteristics	PMGCTB (N = 24)	SMGCTB (N = 26)	*p*-Value
Gender (N, %)			
Male	11 (45.8%)	14 (53.9%)	0.571
Female	13 (54.2%)	12 (46.1%)	
Age (Year, Mean ± SD)	48.0 ± 16.9	43.5 ± 12.8	0.287
Location (N, %)			0.040 *
Limb	6 (25.0%)	5 (19.2%)	
Limb girdle bone	11 (45.9%)	4 (15.4%)	
Axial bone	5 (20.8%)	15 (57.7%)	
Others	2 (8.3%)	2 (7.7%)	
Treatment history (N, %)			<0.001 *
Newly	22 (91.7%)	2 (7.7%)	
Recurrent	2 (8.3%)	24 (92.3%)	
Pathology (N, %)			0.171
OS	6 (25.0%)	12 (46.2%)	
CS	1 (4.2%)	3 (11.5%)	
FS	3 (12.5%)	3 (11.5%)	
UPS	11 (45.8%)	8 (30.8%)	
MFH	3 (12.5%)	0 (0)	
*H3F3A* mutation status (N, %)	n = 22	n = 23	0.020 *
Positive	11 (50.0%)	19 (82.6%)	
Negative	11 (50.0%)	4 (17.4%)	
H3.3G34W (N, %)	n = 22	n = 23	0.300
Positive	10 (45.4%)	14 (60.9%)	
Negative	12 (54.6%)	9 (39.1%)	
H3.3G34V (N, %)	n = 22	n = 23	0.400
Positive	0 (0)	4 (17.4%)	
Negative	22 (100%)	19 (82.6%)	
H3.3G34R (N, %)	n = 22	n = 23	0.577
Positive	1 (4.5%)	2 (8.7%)	
Negative	21 (95.5%)	21 (91.3%)	
Initial metastasis (N, %)			0.600
Yes	1 (4.2%)	2 (7.7%)	
No	23 (95.8%)	24 (92.3%)	
Denosumab (N, %)			0.050 *
Yes	10 (41.7%)	18 (69.2%)	
No	12 (58.3%)	8 (30.8%)	
Chemotherapy (N, %)			0.075
Yes	9 (37.5%)	4 (15.4%)	
No	15 (62.5%)	22 (84.6%)	
Follow-up (Mons, Med/IQR)	38.5 (16–64)	37.5 (13–68)	0.484
Recurrence (N, %)			0.153
Yes	9 (37.5%)	15 (57.7%)	
No	15 (62.5%)	11 (42.3%)	
TTR (Mons, Med/IQR)	8 (3–8)	10 (5–19)	0.150
5-year RFS (95%CI)	62.5% (40.3–78.4%)	36.6% (16.8–56.8%)	0.261
Metastasis (N, %)	n = 23	n = 24	0.260
Yes	6 (26.1%)	10 (41.7%)	
No	17 (73.9%)	14 (58.3%)	
TTM (Mons, Med/IQR)	5 (3–10)	10 (5–21)	0.496
5-year MFS (95%CI)	73.9% (50.1–87.3%)	50.0% (25.2–70.6%)	0.332
Survival (N, %)			0.216
Yes	17 (70.8%)	14 (53.8%)	
No	7 (29.2%)	12 (46.2%)	
TTD (Mons, Med/IQR)	12 (10–16)	12.5 (9.5–20)	1.000
5-year OAS (95%CI)	70.0% (47.0–84.5%)	48.1% (25.5–67.5%)	0.227

PMGCTB: primary malignant giant cell tumor of bone; SMGTCB: secondary malignant giant cell tumor of bone; SD: standard deviation; Med: median; IQR: interquartile interval; OS: osteosarcoma; CS: chondrosarcoma; FS: fibrosarcoma; UPS: undifferentiated pleomorphic sarcoma; MFH: malignant fibrous histiocytoma; TTR: time to recurrence; TTM: time to metastasis; TTD: time to death; RFS: recurrence-free survival; MFS: metastasis-free survival; OAS: overall survival; CI: confidence interval. * means the *p*-Value is significant.

**Table 3 bioengineering-12-00911-t003:** Comparison of the characteristics and prognosis in patients with SMGCTB.

Characteristics	SMGCTB (N = 26)		*p*-Value
	PS (N = 19)	PR (N = 7)	
Gender (N, %)			0.838
Male	10 (52.6%)	4 (57.1%)	
Female	9 (47.4%)	3 (42.9%)	
Age (Year, Mean ± SD)	41.8 ± 14.2	48.0 ± 7.1	0.287
Location (N, %)			0.160
Limb	5 (26.3%)	0 (0)	
Limb girdle bone	9 (47.4%)	6 (85.7%)	
Axial bone	4 (21.0%)	0 (0)	
Others	1 (5.3%)	1 (14.3%)	
Pathology (N, %)			0.983
OS	9 (47.4%)	3 (42.9%)	
CS	2 (10.5%)	1 (14.3%)	
FS	2 (10.5%)	1 (14.3%)	
UPS	6 (31.6%)	2 (28.5%)	
MFH	0 (0)	0 (0)	
LI (Mons, Med/IQR)	31 (7–144)	90 (36–131)	0.140
*H3F3A* mutation status (N, %)	n = 17	n = 6	0.539
Positive	13 (76.5%)	6 (100%)	
Negative	4 (23.5%)	0 (0)	
H3.3G34W (N, %)	n = 17	n = 6	0.340
Positive	9 (52.9%)	5 (83.3%)	
Negative	8 (47.1%)	1 (16.7%)	
H3.3G34V (N, %)	n = 17	n = 6	0.957
Positive	3 (17.7%)	1 (16.7%)	
Negative	14 (82.3%)	5 (83.3%)	
H3.3G34R (N, %)	n = 17	n = 6	0.420
Positive	1 (5.9%)	1 (16.7%)	
Negative	16 (94.1%)	5 (83.3%)	
Initial metastasis (N, %)			0.372
Yes	2 (10.5%)	0 (0)	
No	17 (89.5%)	7 (100%)	
Denosumab (N, %)			0.883
Yes	13 (68.4%)	5 (71.4%)	
No	6 (31.6%)	2 (28.6%)	
Chemotherapy (N, %)			0.925
Yes	3 (15.8%)	1 (14.3%)	
No	16 (84.2%)	6 (85.7%)	
Follow-up (Mons, Med/IQR)	39 (12–79)	25 (13–45)	0.623
Recurrence (N, %)			0.390
Yes	10 (52.6%)	5 (71.4%)	
No	9 (47.4%)	2 (28.6%)	
TTR (Mons, Med/IQR)	11 (5–12)	10 (6–19)	0.711
5-year RFS (95%CI)	45.6% (22.3–66.3%)	38.1% (6.1–71.6%)	0.376
Metastasis (N, %)	n = 17	n = 7	0.939
Yes	7 (41.2%)	3 (42.9%)	
No	10 (58.8%)	4 (57.1%)	
TTM (Mons, Med/IQR)	10 (5–26)	10 (10–21)	0.726
5-year MFS (95%CI)	52.3% (23.3–74.9%)	44.4% (6.6–78.5%)	0.755
Survival (N, %)			0.495
Yes	11 (57.9%)	3 (42.9%)	
No	8 (42.1%)	4 (57.1%)	
TTD (Mons, Med/IQR)	12 (9.5–15)	19 (10.5–35)	0.492
5-year OAS (95%CI)	54.1% (27.4–74.7%)	26.8% (1.3–67.0%)	0.510

PMGCTB: primary malignant giant cell tumor of bone; SMGTCB: secondary malignant giant cell tumor of bone; SD: standard deviation; Med: median; IQR: interquartile interval; LI: latency interval; OS: osteosarcoma; CS: chondrosarcoma; FS: fibrosarcoma; UPS: undifferentiated pleomorphic sarcoma; MFH: malignant fibrous histiocytoma; TTR: time to recurrence; TTM: time to metastasis; TTD: time to death; RFS: recurrence-free survival; MFS: metastasis-free survival; OAS: overall survival; CI: confidence interval.

**Table 4 bioengineering-12-00911-t004:** The outcome of patients with different H3F3A pathogenic variants.

	5-Year RFS (95%CI)	*p*-Value	5-Year MFS (95%CI)	*p*-Value	5-Year OAS (95%CI)	*p*-Value
H3.3G34W		0.163		0.002 *		0.096
Positive	61.1% (35.3–79.2%)		90.7% (67.6–97.6%)		76.4% (51.0–89.8%)	
Negative	37.5% (17.7–57.4%)		28.6% (6.7–55.9%)		47.1% (25.1–66.4%)	
H3.3G34V		0.544		0.224		0.158
Positive	25.0% (8.9–66.5%)		25.0% (8.9–66.5%)		25.0% (8.9–66.5%)	
Negative	52.5% (34.7–67.5%)		66.7% (46.4–80.7%)		66.5% (48.9–79.2%)	
H3.3G34R		0.482		0.199		0.049 *
Positive	33.3% (9.0–77.4%)		33.3% (9.0–77.4%)		33.3% (9.0–77.4%)	
Negative	53.5% (35.8–68.3%)		65.0% (45.3–79.2%)		67.2% (49.9–79.7%)	

RFS: recurrence-free survival; MFS: metastasis-free survival; OAS: overall survival; CI: confidence interval. * means the *p*-Value is siginificant.

**Table 5 bioengineering-12-00911-t005:** Univariate Cox proportional hazard analysis of RFS, MFS, and OS in patients with MGCTB.

Variables	RFS		MFS (N = 47)		OAS	
	HR (95%CI)	*p*-Value	HR (95%CI)	*p*-Value	HR (95%CI)	*p*-Value
Gender		0.333		0.759		0.683
Male	Ref		Ref		Ref	
Female	0.67 (0.30–1.51)		0.86 (0.32–2.29)		0.83 (0.34–2.04)	
Age	1.01 (0.98–1.04)	0.379	1.01 (0.98–1.05)	0.524	1.01 (0.98–1.05)	0.407
Location		0.002 *		0.033 *		0.003 *
Limb	Ref		Ref		Ref	
Limb girdle bone	4.79 (0.58–39.76)	0.147	3.07 (0.34–27.49)	0.316	2.12 (0.22–20.43)	0.515
Axial bone	11.06 (1.44–84.76)	0.021 *	7.75 (0.98–61.45)	0.052	8.20 (1.06–63.61)	0.044 *
Others	56.73 (5.87–548.51)	<0.001 *	15.31 (1.35–173.37)	0.028 *	17.90 (1.98–162.14)	0.010 *
Treatment history		0.228		0.300		0.246
Newly	Ref		Ref		Ref	
Recurrent	1.65 (0.72–3.77)		1.69 (0.62–4.66)		1.72 (0.68–4.37)	
Type		0.267		0.337		0.232
Primary	Ref		Ref		Ref	
Secondary	1.59 (0.69–3.63)		1.63 (0.59–4.50)		1.75 (0.69–4.44)	
Pathology		0.138		0.227		0.433
OS	Ref		Ref		Ref	
CS	1.30 (0.26–6.46)	0.748	2.45 (0.41–14.69)	0.327	1.34 (0.27–6.69)	0.718
FS	1.76 (0.50–6.29)	0.378	2.49 (0.50–12.36)	0.264	0.91 (0.18–4.55)	0.913
UPS	2.15 (0.80–5.75)	0.127	2.74 (0.72–10.37)	0.139	1.37 (0.49–3.86)	0.549
MFH	NA	NA	NA	NA	NA	NA
*H3F3A* mutation status		0.512		0.033 *		0.663
Positive	Ref		Ref		Ref	
Negative	1.35 (0.56–3.26)		3.16 (1.10–9.14)		1.26 (0.46–3.46)	
H3.3G34W		0.068		<0.001 *		0.057
Positive	Ref		Ref		Ref	
Negative	2.25 (0.93–5.45)		8.87 (1.97–40.05)		2.80 (0.97–8.08)	
H3.3G34V		0.373		0.148		0.138
Positive	Ref		Ref		Ref	
Negative	0.55 (0.16–1.88)		0.35 (0.09–1.27)		0.34 (0.09–1.22)	
H3.3G34R		0.061		0.157		0.040 *
Positive	Ref		Ref		Ref	
Negative	0.25 (0.07–0.87)		0.28 (0.06–1.29)		0.26 (0.07–0.94)	
Initial metastasis		0.331	-	-		0.218
Yes	Ref		-	-	Ref	
No	0.45 (0.10–1.92)		-	-	0.35 (0.08–1.52)	
Denosumab		0.919		0.906		0.820
Yes	Ref		Ref		Ref	
No	1.04 (0.46–2.34)		0.94 (0.35–2.54)		1.11 (0.45–2.74)	
Chemotherapy		0.505		0.514		0.974
Yes	Ref		Ref		Ref	
No	0.74 (0.30–1.78)		0.70 (0.24–2.01)		0.98 (0.35–2.74)	

RFS: recurrence-free survival; MFS: metastasis-free survival; OAS: overall survival; CI: confidence interval; OS: osteosarcoma; CS: chondrosarcoma; FS: fibrosarcoma; UPS: undifferentiated pleomorphic sarcoma; MFH: malignant fibrous histiocytoma; Ref: reference. * means the *p*-Value is significant.

## Data Availability

The raw data supporting the conclusions of this article will be made available by the authors upon request.

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
