# Peer review of "Malignant Giant Cell Tumor of Bone: A Study of Clinical, Pathological, and Prognostic Profile from One Single Center"

_bioengineering, 2025, doi:10.3390/bioengineering12090911_

Round 1

Reviewer 1 Report

Comments and Suggestions for Authors

The article "Malignant Giant Cell Tumor of Bone: A Review and Analysis of Clinical, Pathological, and Prognostic Profile from One Single Center" by Shi et al. This article offers valuable insights into malignant giant cell tumors of bone (GCTB), particularly regarding the prognostic significance of tumor location and H3F3A mutation status. The study provides solid clinical evidence that can inform future research and management strategies for this rare and challenging disease.

This study reviews a significant number of cases of malignant giant cell tumors of bone (GCTB), distinguishing between primary malignant giant cell tumors (PMGCTB) and secondary malignant giant cell tumors (SMGCTB). It incorporates H3F3A mutation analysis to provide molecular insights and correlates clinical and molecular features with survival outcomes, including recurrence-free survival (RFS), metastasis-free survival (MFS), and overall survival (OS).

Unlike previous studies, this research uniquely combines histological subtyping, mutation analysis, and survival correlation in a large cohort, offering a novel and clinically relevant contribution to the field.

While the authors of the article presented a sound and in-depth analysis. I have few suggestions for the authors of the manuscript to make the article more readable.

I believe the language is the manuscript should be thoroughly edited for grammar and clarity. some examples of issues

- Change "Limb girdle bone (pelvic)" to "Pelvic bones of the limb girdle." 

- Revise "Denosumab was managed in most of the patients…" to "Denosumab was administered to most patients…"

Abstract 

Clearly state the number of patients diagnosed with PMGCTB (Pediatric Malignant Giant Cell Tumor of Bone) compared to those with SMGCTB (Secondary Malignant Giant Cell Tumor of Bone). Include a brief mention of the statistical methods used and highlight the key significant findings.

Introduction 

More clearly define the significance of malignant transformation in giant cell tumors of bone (GCTB). Provide a stronger justification for focusing on H3F3A mutations, citing relevant prior studies to support this emphasis.

The authors stated in the introduction that the most common substitution in H3F3A involves changing Gly 34 to either tryptophan (Trp) or leucine (Leu). However, they later altered their statement, indicating that the most frequent substitution is G34W, with rare mutations of G34V and G34R mentioned in line 151 of Section 3.1. While the authors discussed valine (Val) and arginine (Arg) mutations, they did not address the leucine mutation.

Results 

Ensure tables and figures include indicators of statistical significance, such as P-values. Expand on the outcomes specific to different mutation types, including G34W, G34R, and G34V.

Discussion 

Investigate possible biological mechanisms that link the G34V mutation to SMGCTB. Additionally, include a discussion on the potential role of denosumab in the process of malignant transformation, acknowledging that this topic is a subject of controversy.

Author Response

Comments 1: Change "Limb girdle bone (pelvic)" to "Pelvic bones of the limb girdle."

Response 1: Thank you for pointing this out. However, the limb girdle bone of tumor not only included pelvic but also scapula in this study, we did not change this expression.

Comments 2: Revise "Denosumab was managed in most of the patients…" to "Denosumab was administered to most patients…"

Response 2: Agree. We have, accordingly, changed this expression. The changes can be found in page 4 line 168-169.

“Denosumab was administered to most patients with SMGCTB compared to PMGCTB (69.2% vs 41.7%, P=0.05).”

Comments 3: Abstract: Clearly state the number of patients diagnosed with PMGCTB (Pediatric Malignant Giant Cell Tumor of Bone) compared to those with SMGCTB (Secondary Malignant Giant Cell Tumor of Bone). Include a brief mention of the statistical methods used and highlight the key significant findings.

Response 3: Agree. We have added the number of patients diagnosed with PMGCTB and SMGCTB as well as the methods for statistical analysis. The changes can be found in page 1 line 29-37.

“The clinical characteristics were comparable … patients with unfavorable prognostic features.”

Comments 4: Introduction: More clearly define the significance of malignant transformation in giant cell tumors of bone (GCTB). Provide a stronger justification for focusing on H3F3A mutations, citing relevant prior studies to support this emphasis. The authors stated in the introduction that the most common substitution in H3F3A involves changing Gly 34 to either tryptophan (Trp) or leucine (Leu). However, they later altered their statement, indicating that the most frequent substitution is G34W, with rare mutations of G34V and G34R mentioned in line 151 of Section 3.1. While the authors discussed valine (Val) and arginine (Arg) mutations, they did not address the leucine mutation.).

Response 4: Agree. We added the significance of malignant transformation of the GCTB. Also, we cited references to support on the emphasis of H3F3A mutations. At last, we deleted the leucine (Leu) in the introduction part. The changes can be found in page 2 line 68-72, page 2 line and page 1 line 47-48.

“Previous studies reported the H3F3A mutation status remained positive in malignant GCTB [27, 28]. Malignant transformation of GCTB can cause great disruption of bone and even lead more aggressive lung metastasis, which result in a relatively poor prognosis, thus it is important to make correct diagnosis to facilitate appropriate treatment.”

Comments 5: Results: Ensure tables and figures include indicators of statistical significance, such as P-values. Expand on the outcomes specific to different mutation types, including G34W, G34R, and G34V.

Response 5: Agree. We added p-values in the figures and tables. A Table including the outcome according to the H3F3A mutation types was also added.

Comments 6: Discussion: Investigate possible biological mechanisms that link the G34V mutation to SMGCTB. Additionally, include a discussion on the potential role of denosumab in the process of malignant transformation, acknowledging that this topic is a subject of controversy.

Response 6: Agree. However, we have not searched studies reporting the same findings of G34V to SMGCTB. We could not investigate the possible mechanisms because this article did not mainly involve the molecular analysis. Future studies could be performed to investigate on such situation. We also are interested in the controversy of the role of denosumab on malignant transformation in GCTB. We added the relevant discussions in this part. The changes can be found in page 14 line 356-362.

“Although in the setting of denosumab therapy, malignant transformation has been sometimes reported to arise [15, 41], this phenomenon is sufficiently uncommon such it remains unclear whether denosumab might be causative [42]. As only 1% patients on denosumab therapy have experienced malignant transformation, similar to the prevalence of malignant transformation in denosumab-naïve tumors. Overall, denosumab is a safe and effective therapy for GCTB.”

Reviewer 2 Report

Comments and Suggestions for Authors

            There is insufficient detail about how missing data and potential misclassification were handled. “For the specimens obtained in the early time, re-stating for potential H3F3A mutation is necessary” but how many samples were restained? how many could not be reassessed? or what impact this may have had on results? Duration, dosage, adherence for denosumab therapy are not described; the paper simply states “how long and how often did the patients take usage of denosumab was available” which is grammatically incorrect and unclear. Vagueness makes readers lose confidence in the conclusions regarding the effect of denosumab.

            The discussion of negative findings, lack of benefit from chemotherapy or denosumab is not sufficiently critical. “No treatment benefits were found…” but authors do not address if the study was powered to detect meaningful differences nor they explore alternative explanations.

            Clinical implications are addressed insufficiently. Although the authors state “we still advocate systemic treatment…for those with poor prognostic features” they do not specify which patient populations might benefit or provide actionable recommendations

            The organization could be improved by more separating results from their interpretation. In the Results section interpretation and suggested explanations (“This may be due to the most patients with pelvic tumors were PMGTCB, and thus wide resection could be performed”) are mixed with data reporting. Adding more subheadings within the Discussion and summarizing key findings at the end of the Results is recommended.

Some claims are not fully supported with citations. Statements regarding the effectiveness and mechanisms of denosumab or the clinical behavior of specific mutations (H3F3A) can benefit from additional primary references.

Comments on the Quality of English Language

The manuscript contains numerous grammatical errors:

  • “a better understand of the characteristics” -> “a better understanding of the characteristics”
  • “how long and how often did the patients take usage of denosumab was available”
  • “GCTB is a benign aggressive bone tumor” -> “GCTB is a locally aggressive benign bone tumor”
  • “No current studies involving a relatively large number of cases have reported the prognosis of patients with malignant GCTB. To accrue enough patients to make cogent treatment recommendations, retrospective studies are needed.” These two sentences are repetitive.

There are places where technical terms and abbreviations are used inconsistently or without sufficient explanation.

Author Response

Comments 1: [There is insufficient detail about how missing data and potential misclassification were handled. “For the specimens obtained in the early time, re-stating for potential H3F3A mutation is necessary” but how many samples were restained? how many could not be reassessed? or what impact this may have had on results? Duration, dosage, adherence for denosumab therapy are not described; the paper simply states “how long and how often did the patients take usage of denosumab was available” which is grammatically incorrect and unclear. Vagueness makes readers lose confidence in the conclusions regarding the effect of denosumab.]

Response 1: Agree. We added the details of re-staining. However, we could not estimate the impact of the data loss on the results. Because this is a retrospective study, we could not get the information of the duration, dosage and adherence of denosumab application and thus we simply divided patients into those with and without denosumab therapy. The detail of re-staining can be found in page 3 line 117-119.

“For the 29 specimens obtained in the early time, re-staining for potential H3F3A pathogenic variant is necessary. However, the staining results were still unavailable for 5 specimens in this cohort.”

Comments 2: [The discussion of negative findings, lack of benefit from chemotherapy or denosumab is not sufficiently critical. “No treatment benefits were found…” but authors do not address if the study was powered to detect meaningful differences nor they explore alternative explanations.]

Response 2: Agree. We added some discussion about the negative findings of systemic therapy. The changes can be found in page 13 line 342-348. Besides we addressed the statistical significance of the differences of chemotherapy and denosumab. The changes can be found in page 5 line 211-212.

“In the current study, no significant survival advantage was observed with chemotherapy. While chemotherapy was applied only in 17 patients, and these patients presented with multiple confounding factors in terms of tumor type, location and treatment history. This made it challenging to directly evaluate the therapeutic role of chemotherapy. Thus, even though our data did not support the efficacy of chemotherapy for malignant GCTB, we should interpret the results cautiously.”

“No statistical treatment benefits were found in the included patients with chemotherapy and denosumab for RFS, MFS and OAS (P>0.05, Figure 3).”

Comments 3: [Clinical implications are addressed insufficiently. Although the authors state “we still advocate systemic treatment…for those with poor prognostic features” they do not specify which patient populations might benefit or provide actionable recommendations.]

Response 3: Agree. We changed the recommendations for chemotherapy application. The changes can be found in page 13 line 349-352

“Since the outstanding results achieved by chemotherapy in other malignant bone tumors, we suggest administration of chemotherapy for selected patients with malignant GCTB, especially for those with inferior prognostic features above mentioned.”

Comments 4: [The organization could be improved by more separating results from their interpretation. In the Results section interpretation and suggested explanations (“This may be due to the most patients with pelvic tumors were PMGTCB, and thus wide resection could be performed”) are mixed with data reporting. Adding more subheadings within the Discussion and summarizing key findings at the end of the Results is recommended.]

Response 4: Agree. We revised the Discussion part. We also added the true data of number of pelvis location in PMGCTB, which was not our data reporting. The changes can be found in page 12 line 306-308. More subheadings were not added, because the other points are not our interests. The key findings were summarizing at the Conclusion part.

“This may be due to the most patients with pelvic tumors was PMGTCB (11/15, 73.3%), and thus wide resection could be performed when compared to SMGCTB which underwent multiple surgeries or radiation.”

Comments 5: [Some claims are not fully supported with citations. Statements regarding the effectiveness and mechanisms of denosumab or the clinical behavior of specific mutations (H3F3A) can benefit from additional primary references.]

Response 5: Agree. We added more primary literatures for supporting the mechanisms of denosumab. However, we could not find the literatures investigating the behavior of the H3F3A mutations in malignant GCTB. We think our study was the first to demonstrate the association between H3F3A mutations and the outcomes.

4. Response to Comments on the Quality of English Language

Point 1: The manuscript contains numerous grammatical errors:

“a better understand of the characteristics” -> “a better understanding of the characteristics”

“how long and how often did the patients take usage of denosumab was available”

“GCTB is a benign aggressive bone tumor” -> “GCTB is a locally aggressive benign bone tumor”

“No current studies involving a relatively large number of cases have reported the prognosis of patients with malignant GCTB. To accrue enough patients to make cogent treatment recommendations, retrospective studies are needed.” These two sentences are repetitive.

There are places where technical terms and abbreviations are used inconsistently or without sufficient explanation.

Response 1: Agree. We have revised the expressions according to the reviewer’s comments.

Reviewer 3 Report

Comments and Suggestions for Authors

Dear Authors,

The article is interesting.

Here are my points:

  1. I suggest replacing “review and analysis” with “study” since this is the actual design
  2. Should you use capital letters for “Musculoskeletal Tumor Center”
  3. It is not clear why the affiliations from 1 to 8 are actually the same.
  4. It is not clear “3.3 % of contemporary” cases. Do you mean all cases from your center?
  5. No need to repeat “in this study” since this is obvious (line 31)
  6. The timeline of the study is mandatory
  7. “Mutation” has been replaced with “pathogenic variant” according to current nomenclature.
  8. Genes name should be in Italics.
  9. Keywords: please add the analyzed genes to attract the readers’ interest
  10. Please avoid repeating terms e.g. “composed of 3 components” (line 43); “can….can” (line 58)
  11. Please explain the abbreviations when first used (even they are well known terms) e.g. RANKL
  12. Lines 75-55. These are not questions, but objectives to be studies amid the analysis
  13. Inclusion criteria should be followed by exclusion criteria.
  14. The tables should be placed next to the main text at the point they are firstly mentioned
  15. Results should start with core data, not with a reference to a table since this is an additional tool
  16. “Newly” patient – do you mean “de novo” or “per primam” diagnosis?
  17. Figure 2 and 3 are very useful and they should be presented at a larger scale within the text. At this point, the legend cannot be read.
  18. Limitations of the study is the last section at Discussion
  19. Please agree that the present work represents a well-established study, and avoid “review” or “series” (as mentioned at Conclusion). The number of patients is important when compare to the general population for this specific ailment.

Well done!

Author Response

Comments 1: [I suggest replacing “review and analysis” with “study” since this is the actual design]

Response 1: Agree. Therefore, we have changed the title.

“[Malignant giant cell tumor of bone: a study of clinical, pathological and prognostic profile from one single center]”

Comments 2: [Should you use capital letters for “Musculoskeletal Tumor Center”.]

Response 2: Agree. We used the capital letters for “Musculoskeletal Tumor Center” in authors’ information.

Comments 3: [It is not clear why the affiliations from 1 to 8 are actually the same.]

Response 3: Thank you for your inquiry. We hereby confirm that all authors of this paper are affiliated with the same research center.

Comments 4: [It is not clear “3.3 % of contemporary” cases. Do you mean all cases from your center?]

Response 4: Yes, we referred to all patients diagnosed with GCTB during the same time period, including both benign and malignant cases.

Comments 5: [No need to repeat “in this study” since this is obvious (line 31).] The changes can be found in page 1 line 31.

Response 5: Agree. We revised the expression.

“The tumor location was mainly at axial and pelvic region (70%) and differed between the two types (P=0.040).”

Comments 6: [The timeline of the study is mandatory]

Response 6: Agree. We added the timeline of the study. The changes can be found in page 1 line 28-29.

“During January 1, 2015 to December 31, 2022, fifty patients were included in the study, which made up of 3.3% of the contemporary GCTB patients.”

Comments 7: [“Mutation” has been replaced with “pathogenic variant” according to current nomenclature.]

Response 7: Agree. We used the “pathogenic variant” instead of the “Mutation” in the Abstract. The changes can be found in page 1 line 32-37 and other expressions appeared later were also revised accordingly.

“H3F3A pathogenic variant presented frequently … for patients with unfavorable prognostic features.”

Comments 8: [Genes name should be in Italics.]

Response 8: Agree. We used Italics for Genes in this paper.

Comments 9: [Keywords: please add the analyzed genes to attract the readers’ interest]

Response 9: Agree. We added the analyzed genes in the Keywords.

Comments 10: [Please avoid repeating terms e.g. “composed of 3 components” (line 43); “can….can” (line 58)]

Response 10: We do not agree. We think this is necessary expression for readers who firstly learn about GCTB, so the expression was retained.

Comments 11: [Please explain the abbreviations when first used (even they are well known terms) e.g. RANKL]

Response 11: Agree. We added the explanation of RANKL followed with abbreviations. The changes can be found in page 2 line 53-56.

“The elucidation of the biology of GCTB led to trails of denosumab in this disease, which is a monoclonal antibody of receptor activator of nuclear factor-κB ligand (RANKL) expressed by the stromal cells.”

Comments 12: [Lines 75-55. These are not questions, but objectives to be studies amid the analysis]

Response 12: Agree. We used the term ”objectives” instead of “questions”.

Comments 13: [Inclusion criteria should be followed by exclusion criteria.]

Response 13: Agree. We added the exclusion criteria after the inclusion criteria. The changes can be found in page 2 line 89-92.

“The exclusion criteria were as follows: 1. patients with diagnosis of giant cell-rich osteosarcoma; 2. patients who received only systemic therapy instead of surgical resection; 3. Patients with incomplete follow-up information or those with follow-up less than 12 months.”

Comments 14: [The tables should be placed next to the main text at the point they are firstly mentioned]

Response 14: Agree. However, the manuscript templates required us to put the tables and figures here. We still thanks for your recommendations for layout revisions.

Comments 15: [Results should start with core data, not with a reference to a table since this is an additional tool]

Response 15: Agree. However, the first part of our results was to display the characteristics of included patients. We think the table could help more intuitively learn about the differences of patients. The core data of this part followed immediately thereafter.

Comments 16: [“Newly” patient – do you mean “de novo” or “per primam” diagnosis?]

Response 16: We referred to the patients who did not receive any surgical treatments.

Comments 17: [Figure 2 and 3 are very useful and they should be presented at a larger scale within the text. At this point, the legend cannot be read.]

Response 17: Agree. We magnified the figures in the paper.

Comments 18: [Limitations of the study is the last section at Discussion]

Response 18: Agree. We relocated the limitation to the end of the Discussion part.

Comments 19: [Please agree that the present work represents a well-established study, and avoid “review” or “series” (as mentioned at Conclusion). The number of patients is important when compare to the general population for this specific ailment.]

Response 19: Agree. We changed the terms “review” to “study” in the Conclusion part. The changes can be found in page 15 line 404.

In our study, we found that the prognosis of malignant GTCB was rather poor, especially for SMGCTB.

Round 2

Reviewer 2 Report

Comments and Suggestions for Authors

The work has now been updated according to my comments. Changes were made. It is now ready for publication

Author Response

Comments 1: [The work has now been updated according to my comments. Changes were made. It is now ready for publication]

Response 1: Thanks a lot for your kind suggestions for our manuscript. 
